# Temporal-spatial trends in childbirth in Ontario, Canada

Steven Habbous[1,2*], John W Snelgrove[3,4], Michaela A Smith[5], Grace Miao[5,6], Alysha Dingwall-Harvey[5], Stephen Petersen[1], Anna Lambrinos[1], David Nguyen[7], Prakesh S Shah[1,3], Erik Hellsten[1]

1 Ontario Health (Strategic Analytics), Toronto, Ontario, Canada, 2 Epidemiology & Biostatistics, Western University, London, Ontario, Canada, 3 Department of Obstetrics & Gynaecology, Division of Maternal-Fetal Medicine, Mt Sinai Hospital, Toronto, Ontario, Canada, 4 Institute of Health Policy, Management & Evaluation, University of Toronto, Toronto, Ontario, Canada, 5 BORN Ontario, Ontario, Canada, 6 CHEO Research Institute, Ontario, Canada, 7 Department of Anesthesia, Scarborough Health Network, Toronto, Ontario, Canada

* Steven.habbous@ontariohealth.ca

## Abstract

### Introduction

The importance of understanding the continuum of care throughout the perinatal/postpartum periods is important for health system monitoring and quality improvement. In this study, we take a broad-ranging and longitudinal perspective to examining long-term changes in obstetric care.

### Methods

This is a retrospective population-based study including all liveborn deliveries from 2010–2023 in Ontario, Canada. We used the hospital Discharge Abstract Database to link delivery and newborn abstracts. We report year-over-year changes in socio-demographics, clinical factors, care patterns, and perinatal and postpartum outcomes.

### Results

The number of in-hospital births decreased from 133,957 in 2010–127,660 in 2023. Over the study period, delivery age increased from a mean 30.6 years (SD 5.5) in 2010 to 32.2 (SD 4.9) in 2023 and there was at least a doubling in the proportion of persons who delivered having preexisting/gestational diabetes (5.6% in 2010, 11.1% in 2023), obesity (1.6% in 2010, 4.6% in 2023), pre-eclampsia/eclampsia (1.1% in 2010, 2.6% in 2023), liver disorders (0.43% in 2010; 1.16% in 2023), and other diseases (4.9% in 2010, 10.7% in 2023), p < 0.0001 for all. The proportion of deliveries performed via C-section increased over time (29.3% in 2010, 34.6% in 2023) but the median length-of-stay decreased 2.68% year-over-year. Use of epidural increased

**Data availability statement:** Data availability: Ontario Health is prohibited from making the data used in this research publicly accessible if it includes potentially identifiable personal health information and/or personal information as defined in Ontario law, specifically the Personal Health Information Protection Act (PHIPA) and the Freedom of Information and Protection of Privacy Act (FIPPA). Due to these legal and ethical restrictions, data will not be made publicly available. However, upon request, data de-identified to a level suitable for public release may be provided (Datarequest@ontariohealth.ca).

**Funding:** The author(s) received no specific funding for this work.

**Competing interests:** The authors have declared that no competing interests exist.

**Abbreviations:** CIHI, Canadian Institute for Health Information; ED, emergency department; DAD, Discharge Abstract Database; ICD-10, International Classification of Diseases, 10th edition; LOS, length of stay; MNCN, maternal-newborn chart number; NACRS, National Ambulatory Care Reporting System; PWD, persons who delivered.

non-linearly over the study period and was less likely at lower-volume hospitals. Although uncommon (<5%), the rate of obstetric trauma and birth trauma increased over the study period, regardless of the mode of delivery (p < 0.0001). Six-month mortality did not change over the study period after delivery, while infant mortality decreased (0.35% in 2010 to 0.26% in 2023). We also observed substantial hospital-level variation in utility of services including midwifery care and access to epidural.

## Conclusion

Over the last 14 years, we found an increasing incidence of people giving birth at an older age and having complicating clinical characteristics at the time of delivery.

## Introduction

The World Health Organization (WHO) announced that the theme of World Health Day 2025 is "Healthy beginnings, hopeful futures" to advocate improving maternal and child healthcare globally [1]. High-income countries like Canada have invested significantly in the global effort to reduce maternal and infant mortality in low/middle-income countries [2]. However, little is known about the current state of obstetrics care in Canada and where quality improvement strategies should be focused. Understanding the continuum of care throughout the perinatal/postpartum periods is important for health system monitoring and quality improvement.

The literature from a Canadian setting appears to be sparse, but evidence suggests increasing rates of comorbidities over time including preeclampsia, gestational diabetes, glucose disorders, and overweight/obesity [3,4]. Apart from the COVID-19 pandemic, we are unaware of any provincial changes in protocols or standards that would have largely influenced perinatal or postpartum outcomes, with the potential exception of the creation of two stand-alone midwifery-led birth centers in Toronto and Ottawa in 2014 [5].

Limited literature has reported that the COVID-19 pandemic impacted these indicators including delayed and reduced prenatal care, fewer post-partum visits, and higher rates of comorbidity during the early pandemic [6]. However, most studies tended to compare perinatal outcomes among pregnant persons who also tested positive for SARS-CoV-2 infection, demonstrating worse outcomes among those testing positive [7] or had follow-up limited to the first year of the pandemic [8].

In this study, we take a broad-ranging and longitudinal perspective to examine long-term changes in the quality of obstetric care in Ontario, representing 40% of all births in Canada [9]. This is particularly useful because understanding how maternal characteristics have changed over time (e.g., age at delivery, presence of comorbidities) may influence medical decision-making and perinatal or postpartum outcomes [10]. We also examine whether these indicators were impacted by the COVID-19 pandemic, extending previous work beyond the first wave [8,11]. These results can be used to identify gaps in knowledge and inform priorities for further exploration, policy development, and data acquisition.

## Methods

### Setting

Ontario provides healthcare under a single-payer system. All in-hospital births are performed on an inpatient basis, and all hospitals in Ontario are mandated to report to the Canadian Institute for Health Information (CIHI) Discharge Abstract Database (DAD) for acute inpatient care. Reporting follows STROBE for observational studies (**S2 File**).

### Cohort creation

The present study includes only Ontario residents who delivered liveborn in any Ontario hospital between January 1, 2010 and December 31, 2023. The cohort was created using the DAD (**Fig 1** for simplified version; **S3 File** for detailed version).

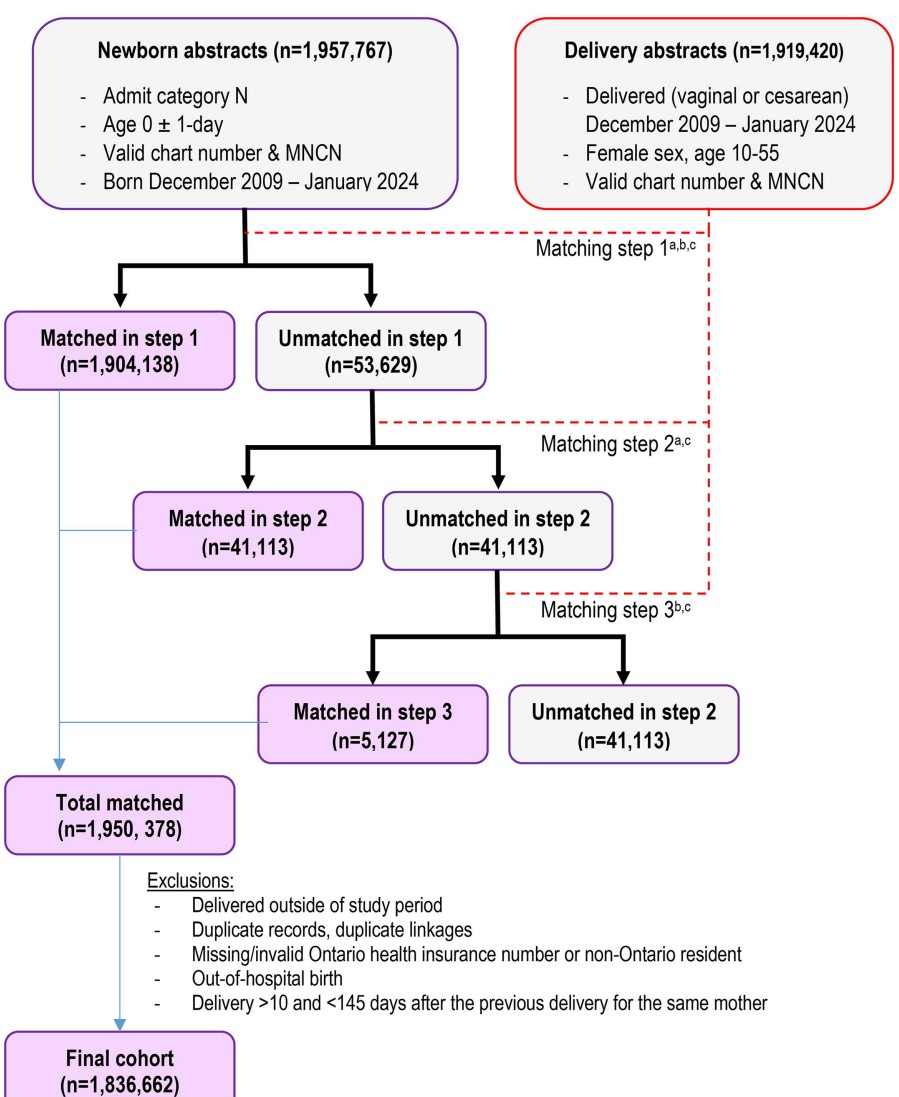

**Fig 1. Simplified cohort creation diagram of liveborn deliveries in Ontario.** Cohort was created using the discharge abstract database (DAD). A detailed cohort creation diagram can be found in the **S3 File**. [a] chart number (person who delivered) = MNCN (newborn). [b] MNCN (person who delivered) = chart number (newborn). [c] admit date (newborn) is within the admit and discharge dates for person who delivered (inclusive). MNCRN – Maternal/Newborn Chart Registration Number.

Using the DAD, we identified all acute inpatient hospital delivery and newborn abstracts with an admit date between December 1, 2009 and January 31, 2024. The 1-month buffer into January 2024 was included to ensure complete linkage (e.g., mothers admitted in December 2023 but delivered in January 2024; a mean 162 newborns were born in the next calendar year as the mothers' admission date). The 1-month buffer into 2009 was to allow all newborns in 2010 to be captured when their mother was admitted in 2009. Data were extracted on June 6, 2024. Linkage across datasets used the healthcard number, a unique patient identifier.

Only liveborn deliveries (vaginal or cesarean) were in scope of the present work. The discharge abstracts were separated into delivery abstracts and newborn abstracts. The cohort was created using ICD10 codes based on the methodology provided by CIHI [12]. To identify a delivery abstract, we looked for women with a diagnosis of childbirth or any other pregnancy-related diagnosis. For this, we searched for the following ICD-10 (International Classification of Diseases, 10th edition) codes in any position (up to 25 diagnostic codes per abstract): Z37 (outcome of delivery); or O10-O16, O21-O46, O48, O60-O75, O85-O92, O95, O98, O99 (ranges are inclusive) with a 6th digit of 1 (delivered) or 2 (delivered with complication) [12]. To identify newborn abstract with a live newborn, abstracts were those with an admission category = "N" (newborn) without a termination code (P964 in any position).

The delivery abstract was required to have female sex coded and age at admission between 10 and 55 inclusive [4]. Throughout, we use the term "person who delivered" (PWD) to describe the individual having the corresponding delivery abstract. We acknowledge that not all those PWD identify as female. Matching required the admission date for the newborn abstract to fall between the admission and discharge dates for the delivery abstract and required the chart number and maternal-newborn chart number (MNCN) of the delivery and newborn abstracts to match (**Fig 1**).

## Matching delivery with newborn abstracts

All abstracts were required to have a non-missing and valid chart number and MNCN, mandatory data elements (**S3 File**). The chart number is unique to an individual at a specific hospital, and all PWD and newborns must each be assigned a unique chart number. The MNCN is also mandatory for all newborn admissions or admissions resulting in a live birth. The MNCN on the abstract for the PWD should match the chart number on the newborns' abstract, and the chart number on the abstract for the PWD should match the MNCN on the newborns' abstract (**Fig 1**; step 1). This applies to singleton births or the first birth of a multiple-birth delivery. For the case of multiple births during the same admission, for non-first-born newborns, the MNCN of the newborns' abstract should match the chart number of the abstract for the PWD, but the converse will not be true because each newborns' chart number must be unique (**Fig 1**; step 2). Lastly, to allow for coding errors, we also matched requiring the MNCN of the abstract for the PWD to match the chart number of the newborns' abstract (**Fig 1**; step 3).

## Exclusions

After matching, we excluded all records where the PWD was not an Ontario resident (responsibility for payment for the admission was not "01", province issuing the healthcard number was not Ontario, or the postal code from the discharge abstract for the delivery was not from Ontario). Out-of-hospital births were excluded, defined as a birth associated with either 1) an admission category = N and an entry code ≠ N; 2) a newborn abstract including any of the following ICD-10 codes: Z381, Z382, Z384, Z385, Z387, Z388; or 3) a delivery abstract (5MD5* or 5MD60) having an out-of-hospital flag = Y. These exclusions were applied last to allow for external validity checks of the matching algorithms with population estimates reported by Statistics Canada (**Figure A in S1 File**) [13]. Out-of-hospital births (whether planned or unplanned) were omitted because the out-of-hospital flags do not represent all out-of-hospital births. The percentage of out-of-hospital births in Ontario is estimated to be 10–20% [14].

## Measures of socio-demographic characteristics

Using the postal code recorded on the discharge abstract at the time of delivery, we captured the PWD's residential neighbourhood-level marginalization derived from the Ontario Marginalization Index using the 2011, 2016, and 2021

Canada Census years for deliveries occurring from 2010–2014, 2015–2019, and 2020–2023, respectively. Due to missing linkage to the 2021 ON-Marg for postal codes for admissions from 2020–2023, we imputed these using the 2016 Census instead. Material resources (formerly material deprivation) includes measures of social welfare and basic living needs (e.g., income quintile, unemployment rate and incomplete high school education rate), while the racialized/newcomer (formerly ethnic diversity) domain is a linear composite of percentage visible minority and immigration within the last 5 years, with most areas showing limited or no changes between census years [15,16].

### Measures of healthcare utilization and delivery during the perinatal period

The perinatal period (or early postpartum period) is generally defined as the time from 22 weeks gestation to 7 days after birth. We measured indicators of healthcare utilization patterns, including epidural use, neonatal intensive care unit utilization, newborn transfers, midwife involvement in the PWD's care and/or delivery, and travel distance for delivery. The PWD's travel distance for delivery was calculated as a straight-line distance using the PWD's postal code on the discharge abstract and the postal code of the institution of delivery. Hospital length of stay was estimated using the date and time of the delivery abstract admission and discharge. For vaginal deliveries, we report the proportion that were instrument-assisted (vacuum, forceps, or both). We measured the rate of caesarean section delivery (C-section) overall and stratified by complexity adapted from the CIHI definition [17]. Pregnancies were considered complex if there was any evidence of multiple gestation, gestational age < 37 or >41 weeks, breech presentation, transverse/oblique lie, multiparity or unknown parity, or evidence of pre-existing/gestational diabetes, pre-existing/gestational hypertension, pre-eclampsia or eclampsia, venous complications, liver disorders, other specified pregnancy-related conditions, anesthesia-related complications during pregnancy, pelvic organ abnormalities, fetal problems, polyhydramnios, oligohydramnios and other amniotic fluid and membrane disorders, placental disorders, placenta previa, placental abruption, antepartum hemorrhage, uterine rupture, obstetric embolism, herpes or HIV infection, or other diseases affecting the PWD [17].

### Outcome measures during the perinatal period

A series of indicators were calculated for each matched abstract pair using well-established and validated indicators (Table A-D in **S1 File**). Using the CIHI Patient Harm indicators methodology, we measured obstetrical hemorrhage, obstetrical trauma (e.g., lacerations, uterine rupture), and birth trauma (injuries to the newborn) [18]. Leveraging the Agency for Healthcare Research and Quality (AHRQ) and the CIHI Hospitalization and Childbirth Quick Stats frameworks, we defined pre-term birth as 22 to <37 weeks gestational age, low-birth-weight births as 500 to <2,500 grams, and small-for-gestational age births as <10th percentile using population-level Canadian estimates [19–24].

### Other measures during the postpartum period

Although less well defined, the postpartum period has been used to refer to the period up to 6 months (or even 1 year) after delivery [25]. We measured unplanned emergency department (ED) visits within 42 days of delivery or birth captured from the CIHI National Ambulatory Care Reporting System (NACRS). We measured maternal and neonatal mortality within 6 months of delivery/birth.

### Other covariates

SARS-CoV-2 infection during the third trimester was determined using evidence of either laboratory evidence (a positive polymerase chain reaction test after linking PWD with the Ontario Laboratory Information System) or the ICD-10 diagnostic code U071 (COVID-19, confirmed) in DAD or NACRS [26]. The timing of the third trimester was derived using the gestational age until week 29.

## Statistics

We used descriptive statistics to report trends over time. All analyses were performed using SAS v9.4 (SAS Institute Inc., Cary, NC). To compute the age-standardized in-hospital births incidence over time while accounting for changes in the age structure of the Ontario population over time, we directly standardized to the 2025 Canada female population using age categories 10–19, 20–24, 25–29, 30–34, 35–39, 40–44, 45–49, 50–55 published by Statistics Canada [27]. For estimating the statistical significance of a trend over time, we used linear regression, logistic regression, or ordinal regression when the outcome was continuous (e.g., length of stay), dichotomous (e.g., C-section delivery), or ordinal (e.g., material resources quintile), respectively. The year of delivery was treated as a continuous variable and only a linear trend was considered. P-values <0.01 were considered statistically significant, but we interpret this cautiously because small changes over time may not be meaningful despite statistical significance owing to the large sample size and the large number of comparisons. Restricted to the most recent time period (2020–2023), we examined hospital-level variability in select measures of healthcare utilization (C-section, midwife involvement, epidural administration) using funnel plots. Funnel plots provide a visual glimpse of proportions while taking into consideration the size of the denominators [28]. We provide a 95% and 99% confidence band for the proportions. Hospitals above (below) the confidence band have a higher (lower) proportion than the mean.

## Privacy

Research ethics was not required as per the Ontario Health privacy assessment as this work was done for the purpose of quality improvement: this study was compliant with section 45(1) of PHIPA (Ontario Health is a prescribed entity). Patient consent was deemed not to be required.

## Results

Between 2010 and 2023, a total 1,845,674/1,957,767 (94.3%) newborn abstracts were matched. When restricted to abstracts for Ontario residents who delivered in-hospital, the linkage rate was higher (96.2%). The number of in-hospital births decreased from 133,957 in 2010–127,660 in 2023 (Table 1). Age-standardized to the Ontario population in 2019, the standardized incidence ratio decreased across all years with the exception of an increase during the first post-pandemic year (2021) (**Fig 2**).

### Sociodemographic characteristics

Over the study period, age at delivery increased from a mean 30.6 years (SD 5.5) in 2010 to 32.2 (SD 4.9) in 2023 (p<0.0001); PWDs were increasingly more likely to reside in areas having higher material resources (15.8% were in the highest quintile in 2010 versus 20.2% in 2023; p<0.0001) (Table 1).

### Clinical characteristics

There was a general trend towards a more complex clinical presentation across a range of indications over the study period. There was at least a doubling in the proportion of PWD having preexisting or gestational diabetes (5.6% in 2010, 11.1% in 2023), obesity (1.6% in 2010, 4.6% in 2023), pre-eclampsia or eclampsia (1.1% in 2010, 2.6% in 2023), some liver disorder (0.43% in 2010; 1.16% in 2023), and other diseases (4.9% in 2010, 10.7% in 2023), p<0.0001 for all (**Fig 3**; Table E in **S1 File**). Despite this, the percentage of PWD categorized as 'complex' increased only marginally from 70.4% in 2010 to 73.8% in 2023 (p<0.0001).

### Care patterns

The hospital length of stay was stable during the study period for vaginal deliveries, exhibiting a year-over-year decrease of 0.93% (median) and 1.16% (75th percentile) (Table F in **S1 File**). For cesarean deliveries, the hospital

**Table 1. Trends of sociodemographic characteristics over time.**

| Socio-demographics | 2010 | 2011 | 2012 | 2013 | 2014 | 2015 | 2016 | 2017 | 2018 | 2019 | 2020 | 2021 | 2022 | 2023 | p |
|---|---|---|---|---|---|---|---|---|---|---|---|---|---|---|---|
| N births | 133,957 | 133,883 | 134,719 | 132,229 | 131,974 | 130,960 | 131,449 | 131,049 | 130,445 | 130,690 | 127,276 | 132,892 | 127,480 | 127,660 | |
| Age at delivery, years | | | | | | | | | | | | | | | *** |
| Median (IQR) | 30.8 (27.0, 34.5) | 30.9 (27.1, 34.5) | 31.0 (27.3, 34.6) | 31.1 (27.5, 34.6) | 31.2 (27.6, 34.6) | 31.3 (27.8, 34.8) | 31.5 (27.9, 34.8) | 31.6 (28.0, 35.0) | 31.7 (28.3, 35.1) | 31.8 (28.5, 35.2) | 32.0 (28.7, 35.3) | 32.1 (28.9, 35.3) | 32.3 (29.1, 35.5) | 32.3 (29.1, 35.5) | |
| Material resources, % | | | | | | | | | | | | | | | *** |
| 1st quintile (highest) | 15.8 | 16.3 | 16.4 | 17.0 | 17.0 | 21.6 | 22.1 | 21.9 | 22.4 | 22.1 | 20.1 | 21.0 | 20.5 | 20.2 | |
| 2 | 18.8 | 18.6 | 18.5 | 18.7 | 18.7 | 19.8 | 19.8 | 19.7 | 19.7 | 20.0 | 20.7 | 21.1 | 21.1 | 20.9 | |
| 3 | 19.1 | 18.9 | 19.2 | 19.2 | 19.2 | 18.4 | 18.5 | 18.4 | 18.4 | 18.6 | 19.8 | 19.8 | 20.0 | 20.2 | |
| 4 | 20.2 | 20.4 | 20.1 | 20.1 | 20.1 | 18.4 | 18.4 | 18.7 | 18.3 | 18.6 | 18.7 | 18.7 | 18.9 | 18.9 | |
| 5th quintile (least) | 26.2 | 25.7 | 25.8 | 24.9 | 25.1 | 21.7 | 21.2 | 21.3 | 21.2 | 20.7 | 20.6 | 19.4 | 19.4 | 19.9 | |
| Racialized/newcomer, % | | | | | | | | | | | | | | | *** |
| 1st quintile (lowest) | 14.0 | 13.7 | 13.6 | 13.8 | 14.0 | 12.8 | 12.9 | 13.1 | 13.2 | 13.2 | 13.9 | 14.3 | 14.1 | 13.7 | |
| 2 | 13.8 | 14.0 | 13.8 | 14.0 | 14.0 | 14.7 | 14.7 | 14.8 | 15.0 | 15.1 | 14.7 | 15.3 | 14.9 | 14.8 | |
| 3 | 16.8 | 16.7 | 16.7 | 16.9 | 17.0 | 17.1 | 16.9 | 17.1 | 17.2 | 17.2 | 16.5 | 17.2 | 16.8 | 16.5 | |
| 4 | 20.4 | 20.4 | 20.1 | 20.6 | 20.4 | 21.3 | 21.2 | 21.0 | 21.2 | 21.1 | 21.4 | 21.3 | 21.5 | 21.5 | |
| 5th quintile (highest) | 35.0 | 35.2 | 35.8 | 34.8 | 34.7 | 34.1 | 34.2 | 34.1 | 33.5 | 33.5 | 33.5 | 31.9 | 32.7 | 33.5 | |

*** p<0.0001

IQR – interquartile range

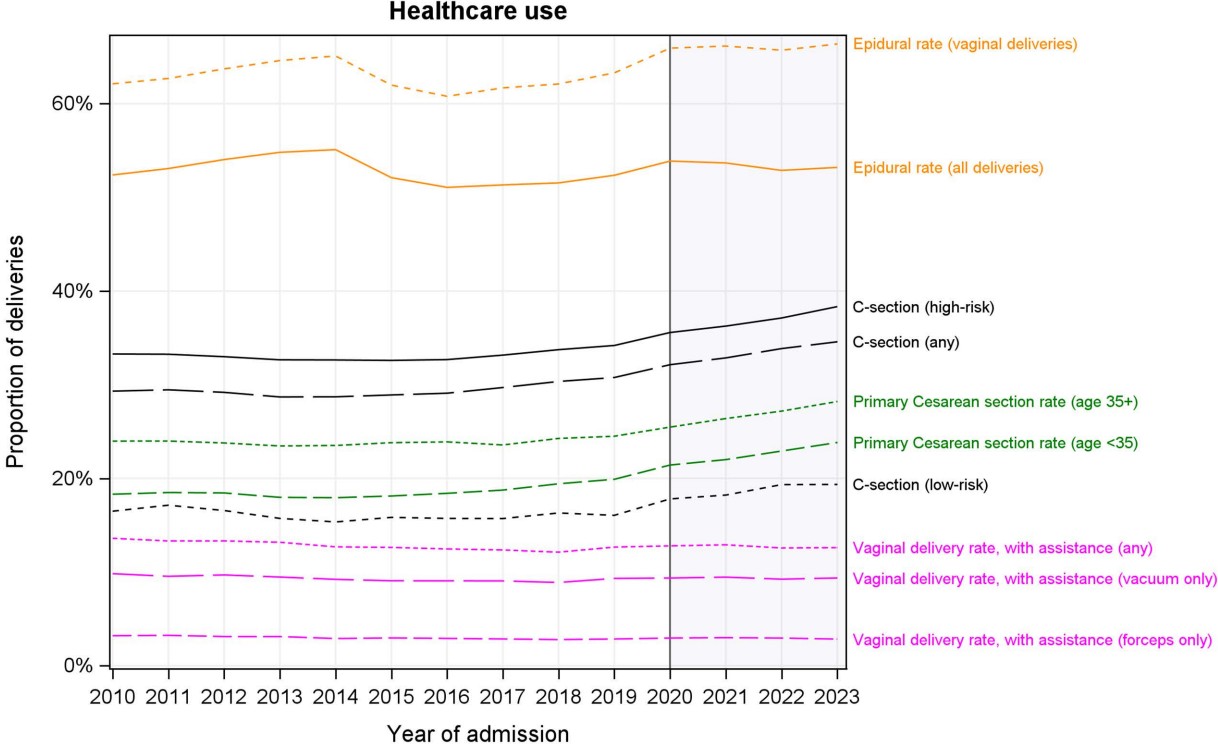

**Fig 2. Age-standardized incidence rate of the number of births performed in-hospital.** Using direct standardization, age-specific rates were applied to the female population in Canada from 2025.

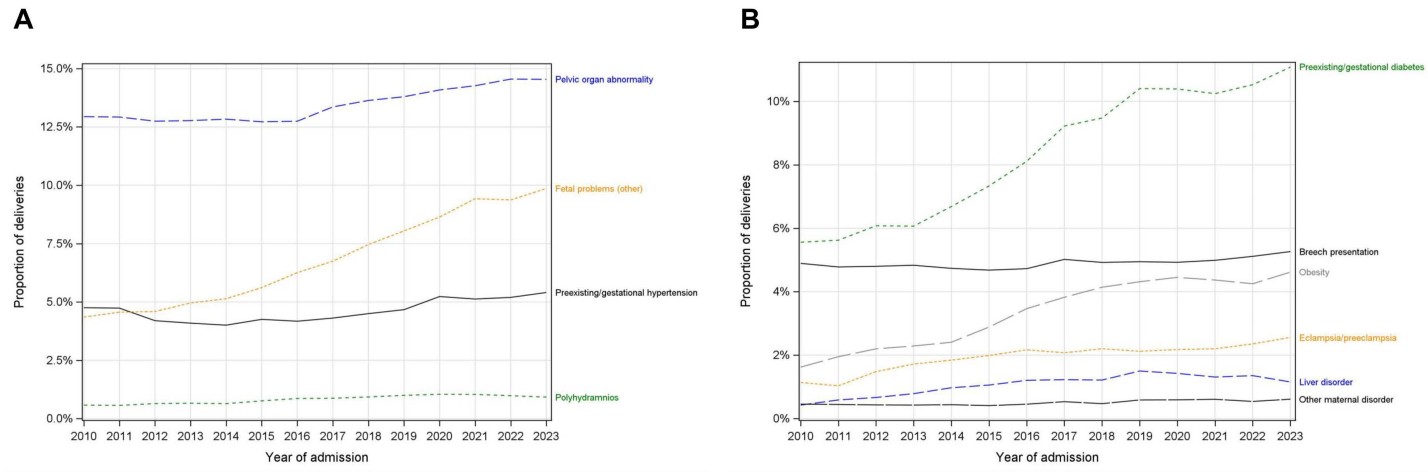

**Fig 3. Incidence of select disorders.** A-B) Percentage of all deliveries where the person who delivered was diagnosed with some complication or condition. Shaded region represents the COVID-19 period.

length of stay (LOS) decreased 2.68% (median) and 2.24% (75th percentile), but this decrease was not stable over the study period (**Fig 4A**).

The use of midwifery care increased steadily from 2010 (7.4%) to 2019 (12.4%) but flattened as of 2020 (start of the COVID-19 pandemic) (**Fig 4B**). An increasing proportion of women travelling >50km was observed over the study period (4.6% in 2010 to 5.9% in 2023). A substantial reduction in ED visits was observed at the onset of the pandemic but appeared to have returned to pre-pandemic levels by 2023. Neonatal intensive care unit utilization and newborn transfer remained steady over the study period with no evidence that the pandemic affected these trends.

Use of epidural exhibited a non-linear pattern: following an increase from 62.1% in 2010 to 65.1% in 2014, then decreased to a nadir of 60.8% in 2016 (**Fig 4C**). The proportion of deliveries performed via C-section increased over time (29.3% in 2010, 34.6% in 2023), an effect that was driven by the COVID-19 pandemic. Between 2010 and 2019, the proportion of deliveries performed by C-section was stable (33% for complex and 15–17% for non-complex deliveries), which increased to 38% and 19% by 2023, respectively (Table F in **S1 File**). Among vaginal births, the proportion of deliveries performed with assistance (vacuum or forceps) remained stable over time.

### Hospital-level variability in outcomes

Hospitals that performed a smaller number of complex deliveries were less likely to deliver via C-section than hospitals performing a greater number of complex deliveries, but no such pattern emerged for non-complex deliveries (**Fig 5A**-**5B**). Pregnant individuals delivering at smaller-volume hospitals were more likely to involve a midwife (**Fig 5C**). Hospitals performing fewer vaginal deliveries were less likely to administer an epidural than hospitals performing more vaginal deliveries (**Fig 5D**).

### Trends in patient outcomes over time

Obstetrical hemorrhage was rare (<1%) and remained stable over the study period (p > 0.02; Table G in **S1 File**). Although uncommon (<5%), the rate of obstetric trauma (PWD) and birth trauma (newborn) increased over the study period, regardless of the mode of delivery (p < 0.0001 for all; **Fig 6A**). Obstetric trauma following instrumented or C-delivery was more common, but also increased over the study period (13.1% in 2010 to 14.7% in 2023). 1-year mortality after delivery increased over the study period for the PWD (p = 0.0004 for overall trend), but infant mortality decreased (34.6 per 10,000 births in 2010 to 25.5 per 10,000 births in 2023) (**Fig 6B**). The proportion of babies born pre-term (gestational age 22–36 weeks), small for gestational age, or low birthweight increased followed a similar pattern, increasing between 2010 and 2019, followed by a lull or decrease in 2020–2021, and increased during 2022 and 2023 (**Fig 6C**).

## Discussion

In this population-based study over a 14-year period, we found an increasing incidence of more complex pregnancies, increased C-section delivery, and substantial hospital-level variation in provision of services.

### General trends

Overall, we found that the risk of perinatal complications (hemorrhage, birth trauma) and postpartum mortality was low, and despite the statistically significant increase over the last 14 years, remained low by 2023. Increasing age at delivery, prevalence of obesity, pre-existing or gestational diabetes, eclampsia/pre-eclampsia, and other maternal disorders may partially be driving these trends.

The optimal rate of C-section delivery has previously been suggested to be between 15% and 19%, a rate exceeded by most high-income nations [29,30]. Internationally, the increasing use of C-section delivery was accelerated during the pandemic [31–37]. We also found a small increase in the use of C-section delivery following the pandemic, regardless of

**A**

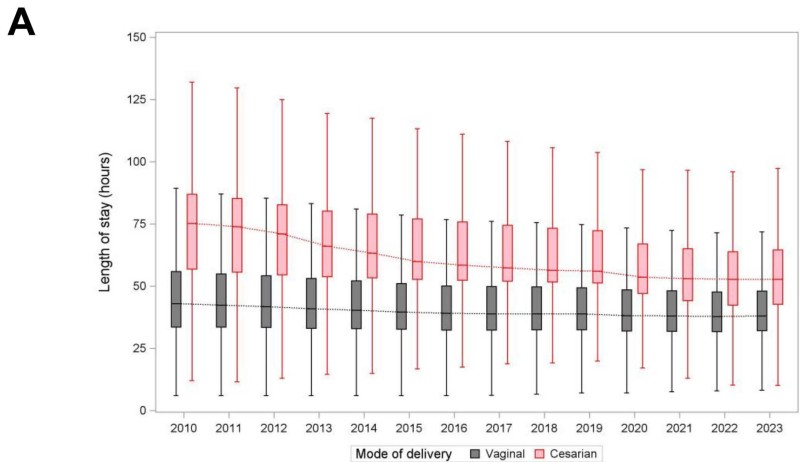

**B**

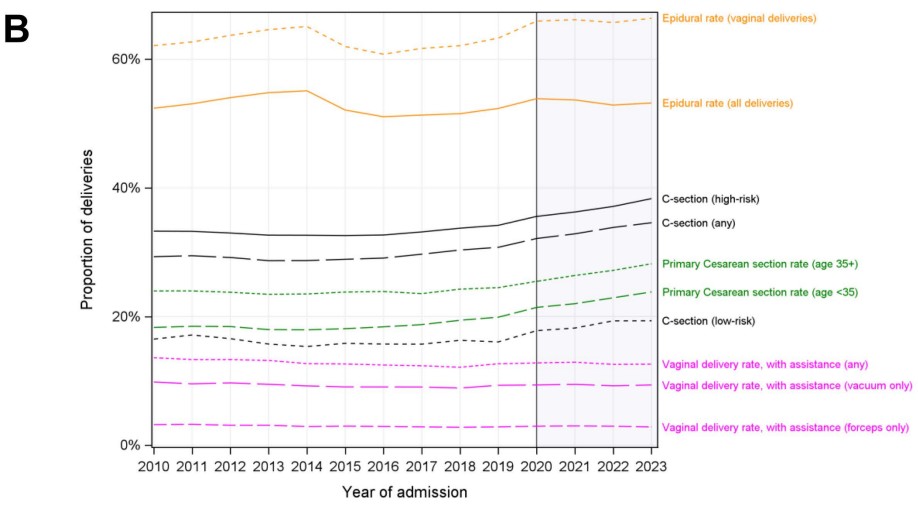

**C**

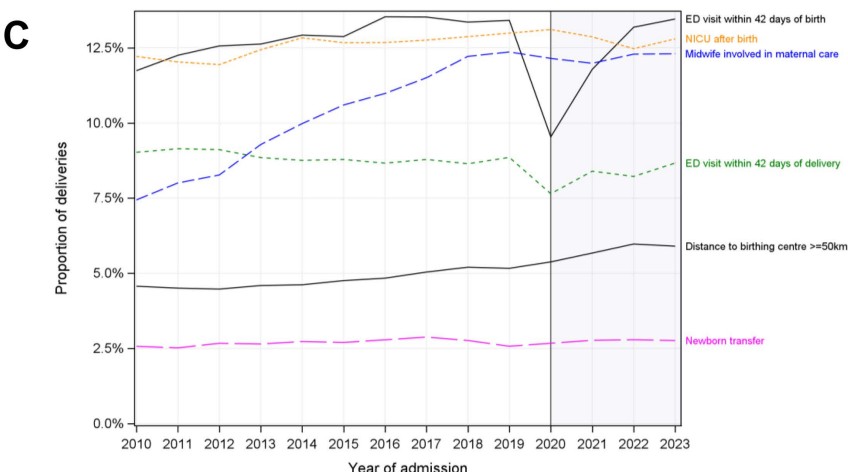

**Fig 4. Measures of healthcare usage and delivery.** A) Total hospital length of stay associated with the person who delivered, by mode of delivery. B) select indicators of access and healthcare utilization. C) select indicators of healthcare delivery. Shaded region represents the COVID-19 period in B-C.

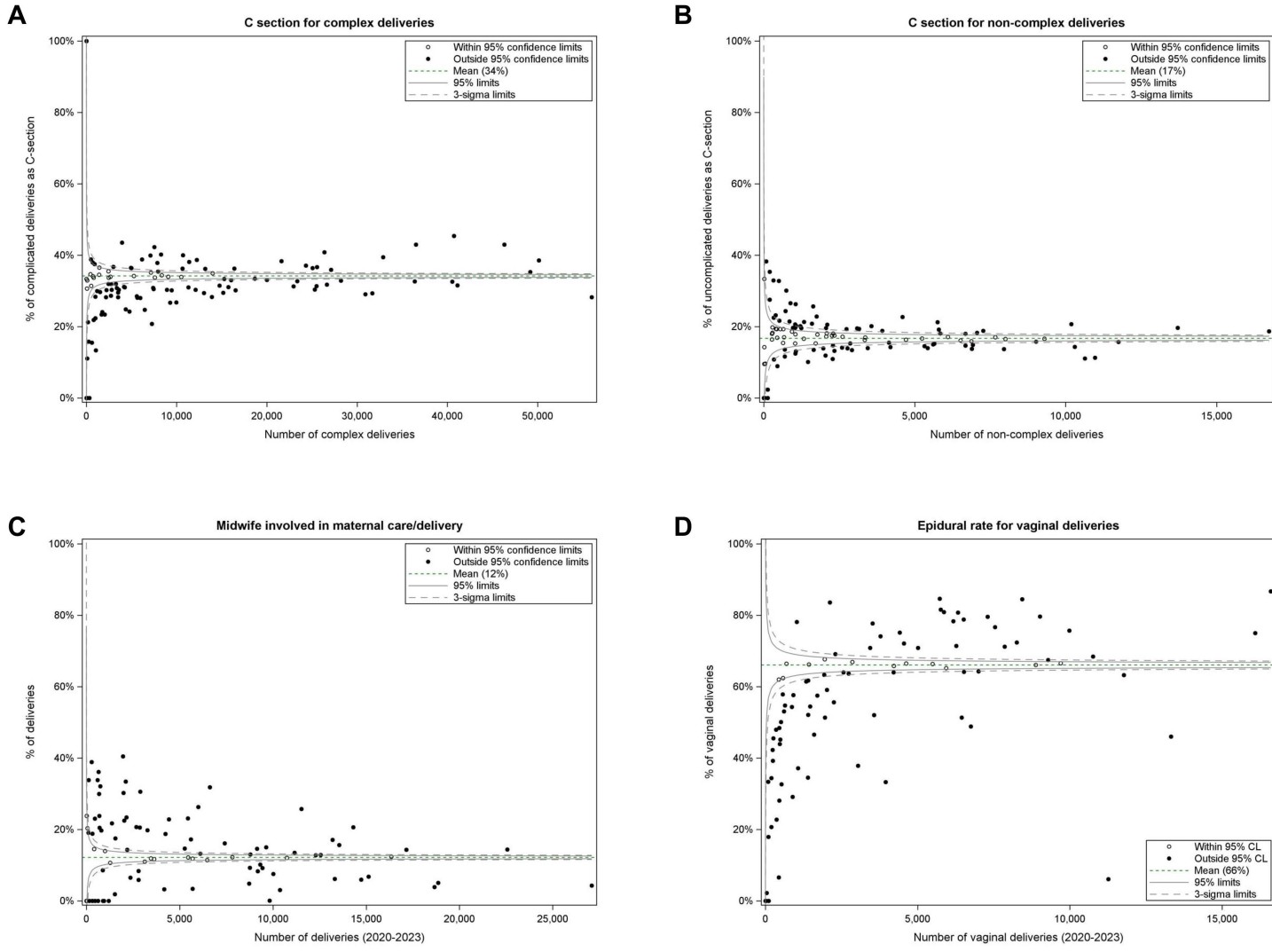

**Fig 5. Funnel plots for hospital-level variation in caesarean section delivery for complex (A) and non-complex (B) deliveries, midwife involvement (C), and epidural use for vaginal deliveries (D).**

complexity categorization. Reasons for these trends are unclear, but may be driven by a rise in unscheduled cesarean delivery associated with increasing maternal age [38]; increasing incidence of maternal or fetal complications such as fetal distress, slow progress in labor, breech presentation, and repeat cesarean section [39–41]; or increasing acceptance among patients and providers that C-section delivery is a normal mode of delivery [42,43]. Hospital culture has also been demonstrated to impact the rate of primary cesarean delivery [44]. Our observation that lower-volume hospitals tended to provide epidurals less frequently or perform fewer C-section deliveries for complex pregnancies supports a regional effect that may be driven by hospital culture, patient preferences, or resource availability. Although the C-section delivery rate is a suboptimal process indicator of healthcare quality on its own, more details are needed on their indications to enable appropriate contextualization [45].

Like our pre-pandemic observations, other studies have reported declining utilization of operative vaginal delivery [46–48]. We observed a small increased use of vacuum-assisted vaginal delivery in 2019 (1 year before the pandemic began), which persisted through the pandemic. Statements released by the Society of Obstetricians and Gynaecologists

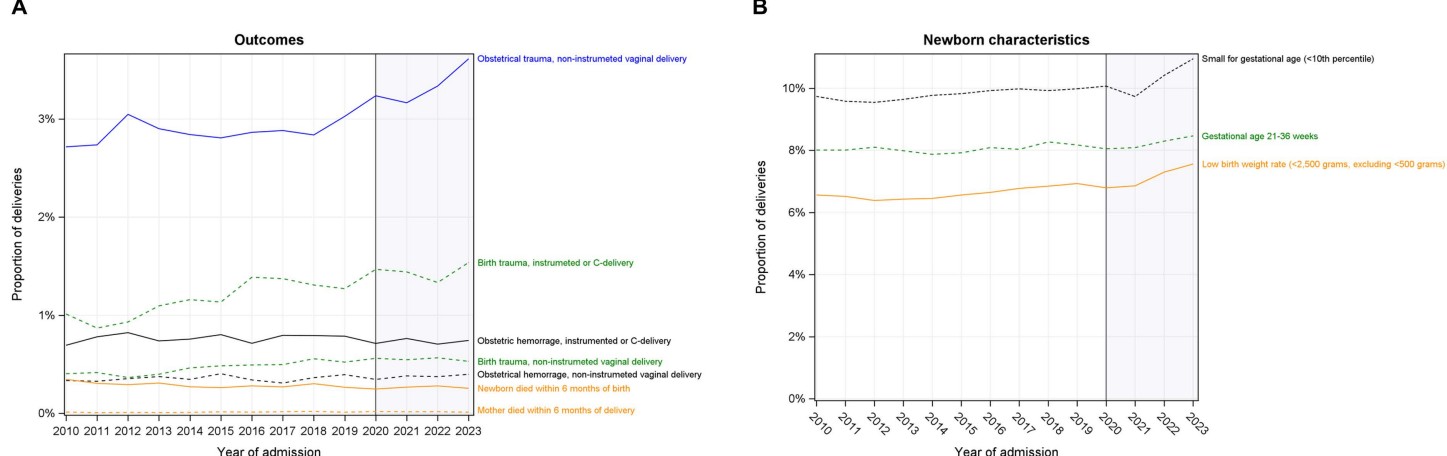

**Fig 6. Measures of outcomes. A)** Rate of obstetric/birth trauma and hemorrhage by mode of delivery over time. **B)** Rate of select newborn indicators over time. Shaded region represents the COVID-19 period.

of Canada (SOGC) in 2019 recommending trials of labor after Cesarean (TOLAC) and reaffirmed during the pandemic may have been a driving force [49,50].

The reduction in LOS observed in Ontario over the last 14 years was also observed internationally before the pandemic [51]. There has been a large push for enhanced recovery after surgery (ERAS) following C-section, which includes improved post-operative analgesia, advancing diet more quickly, and ambulation with one goal being an earlier discharge but also significant reductions in post-operative complications (e.g., deep vein thrombosis) [52]. Despite variation among obstetricians on compliance with different ERAS strategies following C-section delivery, these efforts are believed to be driving the observed reduction in LOS [53]. Considering the mental health, confidence, and wishes of the PWD on a case-by-case basis may also result in safe and effective early discharge [54].

A previous US study reported small increases in complications (obstetric hemorrhage and hypertensive disorders) during the first 18 months of the pandemic, but this was not observed in recent Ontario studies that examined preeclampsia and a composite of multiple severe morbidities for the PWD [37,55]. The higher mortality observed in the US during the COVID-19 era was not mirrored in our study, but the political landscape of obstetrics in the United States likely makes this a poor comparator [37,56]

Hospital-level comparisons can reveal interesting opportunities for potential health system improvement. For example, use of epidural was significantly less likely for PWD at hospitals performing fewer vaginal deliveries. Such hospitals (likely Level 1 facilities) are unlikely to have an anesthesiologist dedicated to labor/delivery, so if the anesthesiologist is otherwise engaged in another procedure, then the window of opportunity for providing an epidural may be missed [57]. There may be other reasons for lower epidural rates at smaller-volume hospitals, including 1) hospital culture and the degree of experience to confidently manage patients who have an epidural in place; and 2) selection bias, such that patients who are more complex and advised to receive an epidural may be preferentially managed at higher-volume centres.

Obstetric unit supply and demand may have shifted since 2020. In Spain, there was a net migration of residents out of the core cities and into rural areas, a large proportion of whom included people of birthing age [58]. Anecdotally, a similar pattern emerged in Ontario, which may partly explain the increase in travel distances in the pandemic observed by 2021 (following a lag required for migration to occur).

Although rare, mortality after delivery increased over the study period while newborn mortality decreased. Most newborn deaths occurring within 1 year of birth occurred within the first 6 months. Reasons for decreasing newborn mortality

may be driven by improvements in prenatal care, uptake of prenatal screening, increasing access to medical abortion that may be differentially used to terminate fetuses having a higher risk of mortality, or improvement in neonatal and infant care [59,60]. In contrast to newborns, deaths occurring within 1 year of delivery were more evenly distributed over time (e.g., approximately half occurred in the first 6 months). Thus, reasons for post-delivery mortality may extend beyond birthing-related complications, including preexisting conditions and post-partum mental health disorders [61–63].

## Limitations

One of the most important limitations is a lack of data on important pregnancy characteristics. For example, C-section rate is no longer used as a measure of quality. However, for instances where a C-section was performed despite achieving full dilatation, this represents a potentially avoidable surgery [64]. Moreover, the quality of administrative data on the pregnant individuals' body mass index remains unclear, but this is an important risk factor for perinatal complications and has increased for obstetric patients over time [65]. Socio-demographic characteristics are also important considerations, but only area-level data were available. Another limitation is the definition of a complex pregnancy. We acknowledge that this is sometimes subjective and may change over time and vary by hospital. Broadly speaking, a complex pregnancy is any pregnancy where there is an elevated risk of adverse outcomes to the newborn or PWD, but what constitute "elevated risk" and "adverse outcomes" are not well defined [66]. The multiparity or unknown parity is "complex" relative to "low-risk pregnancies" by this definition. The multiparity or unknown parity might not be complex pregnancies under other conditions or context. Classifying a pregnancy as complex should have implications on medical decision-making and provision of care (e.g., use of Cesarean delivery; frequency of neonatal blood glucose monitoring).

Internationally, lower birth rates have been reported over the last decade, and Ontario is no exception. At the start of the COVID-19 pandemic, there was a large reduction in the number of in-hospital births that do not appear to be fully accounted for by the reported increase of out-of-hospital births, a trend also seen in the US [37,67]. Moreover, the utilization of midwifery care in the province is underestimated because data were only available for in-hospital births [14]. From hospital administrative databases alone, we are unable to compare outcomes between persons delivering at home (planned or unplanned) with those delivering in-hospital. Another limitation is a lack of information on stillbirths. PWD in-hospital with a stillborn delivery may be at higher risk than those delivering liveborn, but we were only capturing half of the expected number of stillbirths in the province (**S3 File**). For population-based health system monitoring, PWD stillborn should be considered. Another limitation is the potential under-reporting for mortality after birth, although this appears to be small [68].

Another limitation is the lack of data on patient-reported outcome measures and patient-reported experience measures, which can be used to highlight areas for quality improvement beyond medical outcomes [69]. Another limitation of this study is lack of data on health human resources. During the pandemic, hospital staffing shortages caused some hospital units to close temporarily [70]. These tend to be very low-volume centres, so it is difficult to measure the effect on patient outcomes [71]. However, we acknowledge this would be a terrible experience for pregnant/birthing individuals.

There are patients who receive obstetrical care in the province who are not funded by the Ontario Health Insurance Plan (e.g., private payers from out of country; refugees who are covered by various other systems such as the federally funded refugee healthcare program or through midwives uninsured programs). It is conceivable that the demographic and clinical makeup of this population is quite different than the average Ontario population. Further characterization of this population is warranted.

## Conclusion

Over the last 14 years, we found an increasing incidence of deliveries at an older age, increasing incidence of comorbid conditions, higher use of C-section delivery, shorter hospital LOS following C-section delivery, and a reduction in the number of in-hospital births. We also observed substantial hospital-level variation in utility of services including midwifery care, access to epidural, and use of the ED within 42 days after delivery. Access to comprehensive data are needed to encourage quality improvement investigations.

 

## Supporting information

**S1 File. Supporting information.** Administrative codes and definitions of indicators used in this study. Clinical characteristics, healthcare utilization patterns, and outcomes over time.
(DOCX)

**S2 File. STROBE Checklist.** STROBE checklist for observational studies in epidemiology.
(DOCX)

**S3 File. Technical Appendix.** Additional details on data sources and linking newborn abstracts with persons who delivered. Data quality assessments are reported.
(DOCX)

## Acknowledgments

Parts of this material are based on data and information compiled and provided by CIHI. However, the analyses, conclusions, opinions, and statements expressed herein are those of the author, and not necessarily those of CIHI. Parts of this publication are based on data provided by ICES. However, the views expressed in this publication are those of the researcher and do not necessarily represent those of ICES. This report was produced with the support of the Ontario Ministry of Health. However, the views expressed herein are those of the author, and not necessarily those of the Ontario Ministry of Health or the Government of Ontario. We found an increasing incidence of more complex deliveries over time, and hospital-level variation in provision of services in Ontario, a universal healthcare system.

## Author contributions

**Conceptualization:** Steven Habbous, John W Snelgrove, Michaela A Smith, Erik Hellsten.

**Data curation:** Steven Habbous.

**Formal analysis:** Steven Habbous.

**Investigation:** Steven Habbous, John W Snelgrove, Michaela A Smith, Grace Miao, Alysha Dingwall-Harvey, Stephen Petersen, Anna Lambrinos, David Nguyen, Prakesh S Shah, Erik Hellsten.

**Methodology:** Steven Habbous, John W Snelgrove, Michaela A Smith, Grace Miao, Stephen Petersen, Anna Lambrinos, David Nguyen, Prakesh S Shah, Erik Hellsten.

**Supervision:** Erik Hellsten.

**Validation:** Steven Habbous, John W Snelgrove, Michaela A Smith, Grace Miao, Alysha Dingwall-Harvey, Prakesh S Shah.

**Visualization:** Steven Habbous.

**Writing – original draft:** Steven Habbous.

**Writing – review & editing:** Steven Habbous, John W Snelgrove, Michaela A Smith, Grace Miao, Alysha Dingwall-Harvey, Stephen Petersen, Anna Lambrinos, David Nguyen, Prakesh S Shah, Erik Hellsten.

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
