## [Decision Letter · Decision Letter 0]

11 Mar 2025

Dear Dr. Habbous,

Thank you for submitting your manuscript to PLOS ONE. After careful consideration, we feel that it has merit but does not fully meet PLOS ONE’s publication criteria as it currently stands. Therefore, we invite you to submit a revised version of the manuscript that addresses the points raised during the review process.

We look forward to receiving your revised manuscript.

Kind regards,

David Desseauve, MD, MPH, PhD

Academic Editor

PLOS ONE

**Journal Requirements:**

Please ensure that your manuscript meets PLOS ONE's style requirements, including those for file naming. The PLOS ONE style templates can be found at https://journals.plos.org/plosone/s/file?id=wjVg/PLOSOne_formatting_sample_main_body.pdf and https://journals.plos.org/plosone/s/file?id=ba62/PLOSOne_formatting_sample_title_authors_affiliations.pdf 2. We note that you have indicated that there are restrictions to data sharing for this study. For studies involving human research participant data or other sensitive data, we encourage authors to share de-identified or anonymized data. However, when data cannot be publicly shared for ethical reasons, we allow authors to make their data sets available upon request. For information on unacceptable data access restrictions, please see http://journals.plos.org/plosone/s/data-availability#loc-unacceptable-data-access-restrictions.  Before we proceed with your manuscript, please address the following prompts: a) If there are ethical or legal restrictions on sharing a de-identified data set, please explain them in detail (e.g., data contain potentially identifying or sensitive patient information, data are owned by a third-party organization, etc.) and who has imposed them (e.g., a Research Ethics Committee or Institutional Review Board, etc.). Please also provide contact information for a data access committee, ethics committee, or other institutional body to which data requests may be sent. b) If there are no restrictions, please upload the minimal anonymized data set necessary to replicate your study findings to a stable, public repository and provide us with the relevant URLs, DOIs, or accession numbers. Please see http://www.bmj.com/content/340/bmj.c181.long for guidelines on how to de-identify and prepare clinical data for publication. For a list of recommended repositories, please see https://journals.plos.org/plosone/s/recommended-repositories. You also have the option of uploading the data as Supporting Information files, but we would recommend depositing data directly to a data repository if possible. Please update your Data Availability statement in the submission form accordingly.

Reviewers' comments:

Reviewer's Responses to Questions

**Comments to the Author**

1. Is the manuscript technically sound, and do the data support the conclusions?

Reviewer #1: Yes

2. Has the statistical analysis been performed appropriately and rigorously?

Reviewer #1: Yes

3. Have the authors made all data underlying the findings in their manuscript fully available?

Reviewer #1: Yes

4. Is the manuscript presented in an intelligible fashion and written in standard English?

Reviewer #1: Yes

**Reviewer #1:**  Review: temporal spatial trends in labor and delivery in Ontario Review: temporal spatial trends in labor and delivery in Ontario

Thank you for having me review this work that describe temporal-spatial trend in socio-demographics, clinical factors, care patterns and perinatal outcomes over years.

Here are some remarks:

Introduction

1. The overall aim of the study is clear. However, the introduction is very succinct (only 2 references) and, in my opinion, lacks contextual elements.

2. It would be interesting to address the impact of changes in women's socio-demographic and clinical characteristics (e.g. age, obesity, diabetes, hypertensive disorders) on practices (such as caesarean section) and maternal and neonatal morbidity and mortality.

3. You also mentioned assessing the impact of the COVID 19 pandemic in the objective of the study. It would be interesting to give more details on how this pandemic may have affected perinatal health outcomes.

4. Could you also specify whether other work of this type has already been carried out in Canada, and give an estimate of maternal and neonatal mortality? Are any of these indicators collected at national level?

5. Were there any major changes (excluding the COVID 19 pandemic) in perinatal health policies or care services during the study period? If so, please give some contextual information.

Methods

6. What is the proportion of hospital births in Ontario?

7. To clarify the study population from the outset, you may wish to state clearly at the start of the method that you are considering only births to women residing in Ontario during the study period, live births and hospital births. The methodological details come later, and concern the strategy for selecting this population.

8. Please detail the definition of all ICD codes, or consider giving them as appendix

9. It's not clear how the different ICD10 codes were selected. Is it to identify the study population? Is “-“(O10-O16) signify “from O10 to O16”, all selected ? What is the signification of “with a 6th digit of 1 or 2?

10. You might specify first the identification strategy and then the method adopted (for example, to identify delivery-associated abstract, we looked for women with a diagnosis of childbirth or any other pregnancy-related diagnosis. For this, we searched for the following ICD-10 codes…etc). To identify birth-associated abstract with a live newborn, we looked for …etc.). Please consider using the same terms for clarity (newborn abstract / birth associated abstract / birth-associated records – delivery discharge abstract / delivery associated abtract / delivery associated recode).

11. The first three sentences of the “matching” paragraph do not correspond to the matching strategy. You could move them to the previous paragraph in connection with the abstract identification and selection strategy.

12. the “additional inclusions and exclusions” paragraph could be called exclusions only (I don't see any notion of additional inclusions)?

13. I don't understand the sentence (the number of live-born deliveries was comparable, but only before the application of additional exclusions).

14. Regional variability: this part should be on the “statistics” part.

15. Please detail the definition of birth or obstetrical trauma.

16. For indirect standardisation to compute the rate of delivery over time, why did you choose the year 2019 as reference? The denominators were all population or only women? To explain this method, you could point out that it allows to take into account any differences in the age structure of the Ontario population over time. To be more precise, please indicate that you compute the age-standardised in-hospital births incidence ratio (not the rate of delivery over time).

17. Why is multiparity or unknown parity considered as a complex characteristic of pregnancies? I was surprised by the percentage of pregnancies considered complex (around 70%).

18. You fixed your statistical significance set at 1%. Why this choice (I understand that due to the large sample size you had to lowered it but why 1% and not lower?) How was this choice made?

Results

19. Table 1 is very heavy. Couldn't you consider presenting, for example, the most prevalent complexity characteristics in table 1 and the others in an appendix? You could also split it into 2 tables. For continuous variable, consider describe them either as mean(sd) or median(IQR) depending on the distribution (but not both).

20. Could you describe the rate of obstetric haemorrhage overall (not only for assisted and non-assisted vaginal delivery respectively) ?

21. Fig 2: The label for the Y axis (age-standardized in-hospital births incidence ratio) must be specified.

22. Some results are redundant between Table 1 and Figures. It is not useful to present % over time in both figure and table for the same variables. More generally, there are a lot of results and figures. Consider synthesizing to make your message clearer. For example, I wonder whether the impact of the covid 19 pandemic should be addressed in another work.

23. I don't understand the relevance of presenting the covid period in the graphs for maternal characteristics.

24. The first sentence of the regional variability in outcomes paragraph should be presented in the methods section. The healthcare utilization variables analysed should also be detailed in the methods section. You called them regional variability but to me its more hospital level variability as the denominator is the number of deliveries for each hospital.

25. The use of funnel plots and their reading should be further developed. They are not easy to interpret.

26. Why isn't the denominator for funnel plots 5 the total number of deliveries (as for other funnel plots, number of delivery (2020-2023))? If not, the text should specify that it's not just about smaller / higher volume hospitals, but also about those with fewer / higher complex (5A) or non-situations (5B).

27. I'm not sure that figure S2 and these results stratified on SarsCoV2 status are of much interest.

Discussion

28. LOS for length of stay has not been defined previously

29. I'm not convinced by your paragraph discussing the rising C-section rate. If this were essentially explained by the rise in complex pregnancies, you would observe, by stratifying the analyses, an overall increase in caesarean section rates for complex pregnancies, but no increase over time. Are you suggesting that among complex pregnancies, the level of complexity has increased over time? If not, what other factors might explain this increase in C-section rates? Are any strategies being considered to reduce them?

30. You don't mention the reduction in infant mortality over time. Has this been demonstrated previously, and what factors might explain this positive result?

31. Several elements presented in the strengths part of the manuscript are elements of discussion (rate of epidural and travel distance).*

32. Could you looked at emergency vs planned caesarean with your data?

**Do you want your identity to be public for this peer review?** For information about this choice, including consent withdrawal, please see our For information about this choice, including consent withdrawal, please see our Privacy Policy .

Reviewer #1: No

While revising your submission, please upload your figure files to the Preflight Analysis and Conversion Engine (PACE) digital diagnostic tool, https://pacev2.apexcovantage.com/ . PACE helps ensure that figures meet PLOS requirements. To use PACE, you must first register as a user. Registration is free. Then, login and navigate to the UPLOAD tab, where you will find detailed instructions on how to use the tool. If you encounter any issues or have any questions when using PACE, please email PLOS at . PACE helps ensure that figures meet PLOS requirements. To use PACE, you must first register as a user. Registration is free. Then, login and navigate to the UPLOAD tab, where you will find detailed instructions on how to use the tool. If you encounter any issues or have any questions when using PACE, please email PLOS at figures@plos.org . Please note that Supporting Information files do not need this step.. Please note that Supporting Information files do not need this step.

---

## [Author Response · Author response to Decision Letter 1]

25 Apr 2025

Reviewer Comments:

Reviewer #1:

Review: temporal spatial trends in labor and delivery in Ontario

Thank you for having me review this work that describe temporal-spatial trend in socio-demographics, clinical factors, care patterns and perinatal outcomes over years. Here are some remarks:

Introduction

Comment 1: The overall aim of the study is clear. However, the introduction is very succinct (only 2 references) and, in my opinion, lacks contextual elements.

Response: Thank you for the suggestion. We provide additional context in the introduction as it relates to the relevance of the broad approach of the study. We also extended the introduction to align with comments 2 and 3 below.

Comment 2: It would be interesting to address the impact of changes in women's socio-demographic and clinical characteristics (e.g. age, obesity, diabetes, hypertensive disorders) on practices (such as caesarean section) and maternal and neonatal morbidity and mortality.

Response: We added the sentence below to the introduction to frame the importance of the “broad-ranging longitudinal perspective” of the study.

“This is particularly useful because understanding how maternal characteristics have changed over time (e.g., age at delivery, presence of comorbidities) may influence medical decision-making and perinatal outcomes (PMID 39366197). The literature from an Ontario setting appears to be sparse, but Canadian evidence suggests increasing rates of comorbidities over time including preeclampsia, gestational diabetes, glucose disorders, overweight/obesity (37948551, 21184681).”

Comment 3: You also mentioned assessing the impact of the COVID 19 pandemic in the objective of the study. It would be interesting to give more details on how this pandemic may have affected perinatal health outcomes.

Response: We include additional references around the potential impact of the COVID-19 pandemic on 1) healthcare utilization generally; and 2) perinatal outcomes associated with SARS-CoV-2 infection. It reads:

“Previous work has shown delayed and reduced prenatal care, fewer post-partum visits, and higher rates of comorbidity during the early pandemic (PMID 40075540). Most studies tended to compare perinatal outcomes among pregnant persons who also tested positive for SARS-CoV-2 infection, demonstrating worse outcomes among those testing positive (PMID 35282784) or had short follow-up limited to the first year of the pandemic (PMID 34344771).

Comment 4: Could you also specify whether other work of this type has already been carried out in Canada, and give an estimate of maternal and neonatal mortality? Are any of these indicators collected at national level?

Response: We have further reviewed the literature and added references above from a Canadian setting. Several indicators are indeed collected and reported at a national level (https://www.cihi.ca/en/access-data-and-reports/data-tables) and we used these statistics to help validate the linkages we used to measure these indicators. Two key data tables are:

1. Canadian Institute for Health Information. Hospitalization and Childbirth, 1995–1996 to 2023–2024 — Supplementary Statistics. Ottawa, ON: CIHI; 2025.

2. Canadian Institute for Health Information. Inpatient Hospitalization, Surgery and Newborn Statistics, 2023–2024. Ottawa, ON: CIHI; 2025.

Comment 5: Were there any major changes (excluding the COVID 19 pandemic) in perinatal health policies or care services during the study period? If so, please give some contextual information.

Response: This is an interesting question. A review of the literature revealed a few local or small interventions, but not much beyond this. We have added these to the introduction:

“Apart from the COVID-19 pandemic, we are unaware of any provincial changes in protocols or standards that would have largely influenced perinatal outcomes, with the potential exception of the creation of two stand‐alone midwifery‐led birth centers in Toronto and Ottawa in 2014 [9].”

Methods

Comment 6: What is the proportion of hospital births in Ontario?

Response: From our administrative data assets, we cannot reliably estimate the total number of births in Ontario. The out-of-hospital flags do not represent all out-of-hospital births. Using the grey literature, the percentage of home births in Ontario is estimated to be ~15%, decreasing from 19% in the 2010/11 fiscal year to 13% in the 2019/20 fiscal year (https://www.ontariomidwives.ca/midwiferydatamatters-examining-rise-out-hospital-birth-during-covid-19-pandemic.) We added this information to the “Additional Inclusions and exclusions” subsection of the Methods:

“Out-of-hospital births (whether planned or unplanned) were omitted because the out-of-hospital flags do not represent all out-of-hospital births. The percentage of out-of-hospital births in Ontario is estimated to be 10-20% (reference).”

Comment 7: To clarify the study population from the outset, you may wish to state clearly at the start of the method that you are considering only births to women residing in Ontario during the study period, live births and hospital births. The methodological details come later, and concern the strategy for selecting this population.

Response: Thank you for the suggestion. The first sentence of the “Cohort Creation” subsection of the methods now reads:

“The present study includes only Ontario residents who delivered liveborn in an Ontario hospital between 2010 and 2023.”

Comment 8: Please detail the definition of all ICD codes, or consider giving them as appendix

Response: We have added these to supplementary tables S2 (procedure codes) and S3 (ICD10 codes).

Comment 9: It's not clear how the different ICD10 codes were selected. Is it to identify the study population? Is “-“(O10-O16) signify “from O10 to O16”, all selected ? What is the signification of “with a 6th digit of 1 or 2?

Response: In the present revision, in the cohort creation subsection of the methods, we now state “The cohort was created using ICD10 codes based on the methodology provided by CIHI [3].” We now also clarify that the ranges are inclusive (your interpretation is correct), and indicate the significance of the 6th digit of 1 (delivered) and 2 (delivered with complications).

Comment 10: You might specify first the identification strategy and then the method adopted (for example, to identify delivery-associated abstract, we looked for women with a diagnosis of childbirth or any other pregnancy-related diagnosis. For this, we searched for the following ICD-10 codes…etc). To identify birth-associated abstract with a live newborn, we looked for …etc.). Please consider using the same terms for clarity (newborn abstract / birth associated abstract / birth-associated records – delivery discharge abstract / delivery associated abstract / delivery associated recode).

Response: Thank you for the suggestions. We agree that this approach makes it easier to follow. We adapted your wording directly in this part of the methods section. We have gone through the write-up as well to ensure consistency in the use of terminology (we chose to use the terms delivery abstracts and newborn abstracts).

Comment 11: The first three sentences of the “matching” paragraph do not correspond to the matching strategy. You could move them to the previous paragraph in connection with the abstract identification and selection strategy.

Response: We agree and have moved those three paragraphs to the previous subsection of the methods.

Comment 12: the “additional inclusions and exclusions” paragraph could be called exclusions only (I don't see any notion of additional inclusions)?

Response: Thank you for noticing this, we have renamed this subheading to “Exclusions”

Comment 13: I don't understand the sentence (the number of live-born deliveries was comparable, but only before the application of additional exclusions).

Response: I agree this is awkwardly phrased. We removed the part in the parentheses.

Comment 14: Regional variability: this part should be on the “statistics” part.

Response: We agree and have move this to the statistics subsection of the methods.

Comment 15: Please detail the definition of birth or obstetrical trauma.

Response: We now have the detailed descriptions of all the ICD10 codes used in the supplement, so these details are now available. In the methods, we describe obstetrical trauma as including lacerations or uterine rupture. We now also describe birth trauma as injuries to the newborn.

Comment 16: For indirect standardisation to compute the rate of delivery over time, why did you choose the year 2019 as reference? The denominators were all population or only women? To explain this method, you could point out that it allows to take into account any differences in the age structure of the Ontario population over time. To be more precise, please indicate that you compute the age-standardised in-hospital births incidence ratio (not the rate of delivery over time).

Response: Thank you for your question. The choice of 2019 as the reference year was made because it was the most recent year before the pandemic. In the revision, we clarify that the denominators were only women. We have incorporated your suggested wording as well, as it is more accurate.

Comment #17: Why is multiparity or unknown parity considered as a complex characteristic of pregnancies? I was surprised by the percentage of pregnancies considered complex (around 70%).

Response: Thank you for raising this point. The Canadian Institute for health Information defines “low-risk pregnancies” which formed the basis for the definition of “complex”. We acknowledge that this definition of complexity has likely changed over time and may vary from one hospital to the next. The multiparity or unknown parity is “complex” relatively to “low-risk pregnancies” by this definition. The multiparity or unknown parity might not be complex pregnancies under other conditions or context. We have explained it and added the following to the limitations section of the discussion:

“Another limitation is the definition of a complex pregnancy. We acknowledge that this is sometimes subjective and may change over time and vary by hospital. Broadly speaking, a complex pregnancy is any pregnancy where there is an elevated risk of adverse outcomes to the newborn or PWD, but what constitute “elevated risk” and “adverse outcomes” are not well defined (PMID 34175656). The multiparity or unknown parity is “complex” relatively to “low-risk pregnancies” by this definition. The multiparity or unknown parity might not be complex pregnancies under other conditions or context. Classifying a pregnancy as complex should have implications on medical decision-making and provision of care (e.g., use of Cesarean delivery; frequency of neonatal blood glucose monitoring).”

Comment 18: You fixed your statistical significance set at 1%. Why this choice (I understand that due to the large sample size you had to lowered it but why 1% and not lower?) How was this choice made?

Response: Considering the large sample size, we wanted to keep our type 1 error rate to be minimum. The choice of p-value threshold was arbitrary, but 1% is the next most commonly chosen threshold (from experience) less than 5%.

Results

Comment 19: Table 1 is very heavy. Couldn't you consider presenting, for example, the most prevalent complexity characteristics in table 1 and the others in an appendix? You could also split it into 2 tables. For continuous variable, consider describe them either as mean(sd) or median(IQR) depending on the distribution (but not both).

Response: Thank you for the suggestion. We have split the table into four and remove either mean (SD) or median (IQR) where appropriate:

Table 1: Sociodemographic characteristics over time

Table 2: Clinical characteristics over time

Table 3: Healthcare utilization patterns over time

Table 4: Clinical outcomes over time

Comment 20: Could you describe the rate of obstetric haemorrhage overall (not only for assisted and non-assisted vaginal delivery respectively) ?

Response: We have added these statistics. For completeness, we also added this for obstetrical hemorrhage.

Comment 21: Fig 2: The label for the Y axis (age-standardized in-hospital births incidence ratio) must be specified.

Response: We have made this change

Comment 22: Some results are redundant between Table 1 and Figures. It is not useful to present % over time in both figure and table for the same variables. More generally, there are a lot of results and figures. Consider synthesizing to make your message clearer. For example, I wonder whether the impact of the covid 19 pandemic should be addressed in another work.

Response: To strike a balance, we have moved Table 2 (Clinical characteristics over time), Table 3 (Healthcare utilization patterns over time), and Table 4 (Clinical outcomes over time) to the supplement. We find visualizations are a more efficient means to convey information, but interested readers can refer to the appendix for the actual estimates. We have opted to retain the results related to the COVID-19 pandemic because this was an impactful aspect affecting temporal changes in healthcare utilization patterns (and possibly outcomes).

Comment 23: I don't understand the relevance of presenting the covid period in the graphs for maternal characteristics.

Response: Agreed, we have removed this aspect of Figure 3

Comment 24: The first sentence of the regional variability in outcomes paragraph should be presented in the methods section. The healthcare utilization variables analysed should also be detailed in the methods section. You called them regional variability but to me its more hospital level variability as the denominator is the number of deliveries for each hospital.

Response: We have moved this sentence to the methods section (combined with comment #14 above) and have renamed it as “hospital level variability” instead of “regional variability” throughout the manuscript.

Comment #25: The use of funnel plots and their reading should be further developed. They are not easy to interpret.

Response: In the methods section, we have added interpretative advice for the funnel plots and references as well:

“Restricted to the most recent time period (2020-2023), we examined hospital-level variability in select measures of healthcare utilization (C-section, midwife involvement, epidural administration) using funnel plots. Funnel plots provide a visual glimpse of proportions while taking into consideration the size of the denominators [28]. We provide a 95% and 99% confidence band for the proportions. Hospitals above (below) the confidence band have a higher (lower) proportion than the population mean for a hospital with its specific volume of deliveries.”

Comment 26: Why isn't the denominator for funnel plots 5 the total number of deliveries (as for other funnel plots, number of delivery (2020-2023))? If not, the text should specify that it's not just about smaller / higher volume hospitals, but also about those with fewer / higher complex (5A) or non-situations (5B).

Response: Thank you for raising this. The x-axis for the funnel plot corresponds to the denominator of the indicator, so they are different for each funnel plot. Your interpretation is correct, and we have re-worded the results section to read as follows:

“Hospitals that performed a smaller number of complex deliveries were less likely to deliver via C-section than hospitals performing a greater number of complex deliveries, but no such pattern emerged for non-complex deliveries (Fig 5A-B). Pregnant individuals delivering at smaller-volume hospitals were more likely to involve a midwife (Fig 5C). Hospitals performing fewer vaginal deliveries were less likely to administer an epidural than hospitals performing more vaginal deliveries (Fig 5D).”

Comment #27: I'm not sure that figure S2 and these results stratified on SarsCoV2 status are of much interest.

Response: We agree and have removed this supplementary figure and its reference from the paper.

Discussion

Comment #28: LOS for length of stay has not been defined previously

Response: Thank you for notic

---

## [Editor Report · Decision Letter 1]

5 Jun 2025

Dear Dr. Habbous,

Thank you for submitting your manuscript to PLOS ONE. After careful consideration, we feel that it has merit but does not fully meet PLOS ONE’s publication criteria as it currently stands. Therefore, we invite you to submit a revised version of the manuscript that addresses the points raised during the review process.

We look forward to receiving your revised manuscript.

Kind regards,

David Desseauve, MD, MPH, PhD

Academic Editor

PLOS ONE

Journal Requirements:

**Additional Editor Comments:**

I find that the authors have addressed the majority of the reviewer comments with care and diligence. The introduction has been meaningfully expanded to provide a stronger contextual foundation, and the methods section now includes sufficient technical detail and transparency regarding cohort creation, data linkage, and indicator definitions. The restructuring of tables and streamlining of figures improve clarity, and the decision to relocate more granular data to the supplement is appropriate.

Moreover, the discussion appropriately reflects on the policy and clinical relevance of the findings while acknowledging the inherent limitations of administrative data.

While a few minor clarifications could further strengthen the manuscript (e.g., deeper discussion of alternative explanations for the rise in cesarean delivery or stratified outcomes by complexity subgroups)

While revising your submission, please upload your figure files to the Preflight Analysis and Conversion Engine (PACE) digital diagnostic tool, https://pacev2.apexcovantage.com/ . PACE helps ensure that figures meet PLOS requirements. To use PACE, you must first register as a user. Registration is free. Then, login and navigate to the UPLOAD tab, where you will find detailed instructions on how to use the tool. If you encounter any issues or have any questions when using PACE, please email PLOS at . PACE helps ensure that figures meet PLOS requirements. To use PACE, you must first register as a user. Registration is free. Then, login and navigate to the UPLOAD tab, where you will find detailed instructions on how to use the tool. If you encounter any issues or have any questions when using PACE, please email PLOS at figures@plos.org . Please note that Supporting Information files do not need this step.

---

## [Author Response · Author response to Decision Letter 2]

23 Jun 2025

Additional Editor Comments:

I find that the authors have addressed the majority of the reviewer comments with care and diligence. The introduction has been meaningfully expanded to provide a stronger contextual foundation, and the methods section now includes sufficient technical detail and transparency regarding cohort creation, data linkage, and indicator definitions. The restructuring of tables and streamlining of figures improve clarity, and the decision to relocate more granular data to the supplement is appropriate.

Moreover, the discussion appropriately reflects on the policy and clinical relevance of the findings while acknowledging the inherent limitations of administrative data.

Comment: While a few minor clarifications could further strengthen the manuscript (e.g., deeper discussion of alternative explanations for the rise in cesarean delivery or stratified outcomes by complexity subgroups)

Response: Thank you for the suggestion. We have added additional thoughts and references to the discussion to potentially explain the observed rise in cesarean delivery over time.

The optimal rate of C-section delivery has previously been suggested to be between 15% and 19%, a rate exceeded by most high-income nations [27,28]. Internationally, the increasing use of C-section delivery was accelerated during the pandemic [29–35]. We also found a small increase in the use of C-section delivery following the pandemic, regardless of complexity categorization. Reasons for these trends are unclear, but may be driven by a rise in unscheduled cesarean delivery associated with increasing maternal age (PMID 37062508); increasing incidence of maternal or fetal complications such as fetal distress, slow progress in labor, breech presentation, and repeat cesarean section (PMC4126949, 34971814, 31976544, 34971814); or increasing acceptance among patients and providers that C-section delivery is a normal mode of delivery (PMC9049329, 37216058). Hospital culture has also been demonstrated to impact the rate of primary cesarean delivery (PMC6407356). Our observation that lower-volume hospitals tended to provide epidurals less frequently or perform fewer C-section deliveries for complex pregnancies supports a regional effect that may be driven by hospital culture, patient preferences, or resource availability. Although the C-section delivery rate is a suboptimal process indicator of healthcare quality on its own, more details are needed on their indications to enable appropriate contextualization [36].

---

## [Decision Letter · Decision Letter 2]

12 Feb 2026

Dear Dr. Habbous,

We look forward to receiving your revised manuscript.

Kind regards,

Emma Campbell, Ph.D

Staff Editor

PLOS One

Journal Requirements:

Reviewers' comments:

Reviewer's Responses to Questions

**Comments to the Author**

Reviewer #1: All comments have been addressed

Reviewer #2: (No Response)

2. Is the manuscript technically sound, and do the data support the conclusions?

Reviewer #1: Yes

Reviewer #2: Partly

3. Has the statistical analysis been performed appropriately and rigorously?

Reviewer #1: Yes

Reviewer #2: Yes

4. Have the authors made all data underlying the findings in their manuscript fully available?

Reviewer #1: Yes

Reviewer #2: Yes

5. Is the manuscript presented in an intelligible fashion and written in standard English?

Reviewer #1: Yes

Reviewer #2: Yes

Reviewer #1: The authors have fully addressed the reviewers' and editor's requests to improve their manuscript for publication.

Reviewer #2: Many thanks for the opportunity to review the manuscript “Temporal-spatial trends in labor and delivery in Ontario,” which addresses an important topic in the current political climate. However, the study may be of interest to a relatively limited group of stakeholders.

The title does not appear to align with the manuscript’s content. I could not identify any data related to “labor,” as the analysis focuses on “delivery.” Using the term “childbirth” may provide a more accurate description.

From the abstract it reads: Conclusion: Over the last 14 years, we found an increasing incidence of people giving birth having more complex demographic and clinical characteristics. How do the authors define “complex demographic”? And which data are this statement based on?

It reads in the manuscript “Understanding the continuum of care throughout the perinatal period is important for health system monitoring and quality improvement”. Perinatal period standard definition is from 22 completed weeks of gestation (154 days) to 7 completed days after birth but the authors go beyond this period. Why or do the authors define the perinatal period differently?

Why did the study only include liveborn deliveries (vaginal or caesarean) and not considering stillbirths? It would be very important to include stillbirth as well.

Please add a reference to this statement: In this study, we take a broad-ranging and longitudinal perspective to examine long-term changes in the quality of obstetric care in Ontario, representing 40% of all births in Canada.

Why using the cut-off of 6 months mortality (from the manuscript: We measured maternal and neonatal mortality within 6 months of delivery/birth) as the authors are interested in the perinatal period according to the manuscript?

The authors report on unplanned emergency department (ED) visits within 42 days of delivery or birth. Why 42 days if the focus of the analysis is during the perinatal period? I find it a bit confusing using “….visits within 42 days of delivery or birth” instead of saying postpartum.

Not sure I understand this paragraph “Funnel plots provide a visual glimpse of proportions while taking into consideration the size of the denominators [26]. We provide a 95% and 99% confidence band for the proportions. Hospitals above (below) the confidence band have a higher (lower) proportion than the population mean for a hospital with its specific volume of deliveries”.

Please add labels to Table 1 Table: Trends of sociodemographic characteristics over time. Not clear for the reader to understand how Material resources are defined as well as Racialized/newcomer.

Fig 2: Age-standardized incidence ratio of the number of births. Rates were indirectly standardized to the rate in 2019. Is this ratio or rate?

Why did the authors not apply the Robson Classification (https://robson-classification-platform.srhr.org/about)?

The authors write: Access to comprehensive data are needed to encourage quality improvement investigations. Suppose not only a matter of access, are the data collected?

Please provide an abbreviation list.

**Do you want your identity to be public for this peer review?** For information about this choice, including consent withdrawal, please see our For information about this choice, including consent withdrawal, please see our Privacy Policy .

Reviewer #1: **Yes:** Pauline Blanc-PetitjeanPauline Blanc-Petitjean

Reviewer #2: **Yes:** Ann-Beth MollerAnn-Beth Moller

---

## [Author Response · Author response to Decision Letter 3]

13 Feb 2026

February 13, 2026

Re: PONE-D-24-52959.R2 “Temporal-spatial trends in labor and delivery in Ontario”

Dear Dr Campbell,

On behalf of all authors, we thank you and the reviewers for the review and the additional suggestions to improve the manuscript. On behalf of all authors, we thank you for your consideration.

Sincerely,

Steven Habbous, PhD

Epidemiologist

Email: shabbous@uwo.ca; steven.habbous@ontariohealth.ca

Reviewer Comments

Reviewer #1:

Comment #1: The authors have fully addressed the reviewers' and editor's requests to improve their manuscript for publication.

Response: Thank you.

Reviewer #2:

Reviewer #2: Many thanks for the opportunity to review the manuscript “Temporal-spatial trends in labor and delivery in Ontario,” which addresses an important topic in the current political climate. However, the study may be of interest to a relatively limited group of stakeholders.

Comment #1: The title does not appear to align with the manuscript’s content. I could not identify any data related to “labor,” as the analysis focuses on “delivery.” Using the term “childbirth” may provide a more accurate description.

Response: Thank you for the suggestion, we have retitled the manuscript to read “Temporal-spatial trends in childbirth in Ontario”.

Comment #2: From the abstract it reads: Conclusion: Over the last 14 years, we found an increasing incidence of people giving birth having more complex demographic and clinical characteristics. How do the authors define “complex demographic”? And which data are this statement based on?

Response: Thank you for noticing this, we agree that this is unclear. The part about demographics relates to age, but the part about clinical characteristics refers to conditions like gestational diabetes, obesity, pre-eclampsia, etc. We rephrased the conclusion to read: “Over the last 14 years, we found an increasing incidence of people giving birth at an older age and having complicating clinical characteristics at the time of delivery.”

Comment #3: It reads in the manuscript “Understanding the continuum of care throughout the perinatal period is important for health system monitoring and quality improvement”. Perinatal period standard definition is from 22 completed weeks of gestation (154 days) to 7 completed days after birth but the authors go beyond this period. Why or do the authors define the perinatal period differently?

Response: Thank you for pointing this out. As we delve further into this space, it would be important to clearly articulate and define the time-periods used, although there is some heterogeneity in the literature particularly around the post-partum period (PMC11051636). In this revision, we introduce the term postpartum and try to avoid using the terms perinatal and postpartum interchangeably. In the methods, we describe measures of healthcare utilization and delivery and outcomes specific to the perinatal period.

Comment #4: Why did the study only include liveborn deliveries (vaginal or caesarean) and not considering stillbirths? It would be very important to include stillbirth as well.

Response: We agree that this is a limitation and is a current methodological issue that we are keen to resolve. The limitations section of the discussion reads as follows:

“Another limitation is a lack of information on stillbirths. PWD in-hospital with a stillborn delivery may be at higher risk than those delivering liveborn, but we were only capturing half of the expected number of stillbirths in the province (Technical Appendix). For population-based health system monitoring, PWD stillborn should be considered.”

Comment #5: Please add a reference to this statement: In this study, we take a broad-ranging and longitudinal perspective to examine long-term changes in the quality of obstetric care in Ontario, representing 40% of all births in Canada.

Response: We have added the following reference:

Statistics Canada. Table 13-10-0414-01 Live births, by place of residence of mother

DOI: https://doi.org/10.25318/1310041401-eng

Comment #6: Why using the cut-off of 6 months mortality (from the manuscript: We measured maternal and neonatal mortality within 6 months of delivery/birth) as the authors are interested in the perinatal period according to the manuscript? The authors report on unplanned emergency department (ED) visits within 42 days of delivery or birth. Why 42 days if the focus of the analysis is during the perinatal period? I find it a bit confusing using “….visits within 42 days of delivery or birth” instead of saying postpartum.

Response: In the present revision we have made the distinction between the perinatal period and the postpartum period. The literature also seems to be inconsistent on whether to report mortality and ED visits at 42 days, 6 months, or 1 year (we have found all three), so we follow suit to be consistent with this.

Comment #7: Not sure I understand this paragraph “Funnel plots provide a visual glimpse of proportions while taking into consideration the size of the denominators [26]. We provide a 95% and 99% confidence band for the proportions. Hospitals above (below) the confidence band have a higher (lower) proportion than the population mean for a hospital with its specific volume of deliveries”.

Response: We have simplified this description to read as follows:

“Funnel plots provide a visual glimpse of proportions while taking into consideration the size of the denominators [26]. We provide a 95% and 99% confidence band for the proportions. Hospitals above (below) the confidence band have a higher (lower) proportion than the population mean for a hospital with its specific volume of deliveriesmean”.

Comment #8: Please add labels to Table 1 Table: Trends of sociodemographic characteristics over time. Not clear for the reader to understand how Material resources are defined as well as Racialized/newcomer.

Response: Thank you, we have indicated that these are quintiles

Comment #9: Fig 2: Age-standardized incidence ratio of the number of births. Rates were indirectly standardized to the rate in 2019. Is this ratio or rate?

Response: Thank you for your question. We plotted age-standardized incidence ratios (although the rates are implicitly used to compute them). To make the estimates more useful to other readers who wish to compare their own rates to the same reference population, we switched to direct standardization using the 2025 female population. The new figure looks similar to the previous one, except the y-axis is now a rate instead of a ratio and the legend is updated:

Fig 2: Age-standardized incidence rate of the number of births performed in-hospital. Using direct standardization, age-specific rates were applied to the female population in Canada from 2025.

Comment #10: Why did the authors not apply the Robson Classification (https://robson-classification-platform.srhr.org/about)?

Response: Thank you for the question. We did not want to comment on the appropriateness of C-section delivery and the Robson score was not available in our data until 2017. This should be the focus of a future study, and we await some more detailed information as our data assets mature (see next comment).

Comment #11: The authors write: Access to comprehensive data are needed to encourage quality improvement investigations. Suppose not only a matter of access, are the data collected?

Response: We believe much of the data are collected by BORN (Better Outcomes Registry & Network). However, we do not (yet) have access to these data and are hoping to be able to access it for future linkage and more in-depth analysis.

Comment #12: Please provide an abbreviation list.

Response: We have added this and reduced the use of abbreviations throughout.

Thank you for all your suggestions and for taking the time to review!

---

## [Decision Letter · Decision Letter 3]

9 Mar 2026

Temporal-spatial trends in childbirth in Ontario, Canada

PONE-D-24-52959R3

Dear Dr. Steven Habbous,

We’re pleased to inform you that your manuscript has been judged scientifically suitable for publication and will be formally accepted for publication once it meets all outstanding technical requirements.

Kind regards,

Hale Teka, MD, MSc, Associate Professor of OBGYN

Academic Editor

PLOS One

Additional Editor Comments (optional):

Reviewers' comments:

Reviewer's Responses to Questions

**Comments to the Author**

Reviewer #2: All comments have been addressed

2. Is the manuscript technically sound, and do the data support the conclusions?

Reviewer #2: Yes

3. Has the statistical analysis been performed appropriately and rigorously?

Reviewer #2: Yes

4. Have the authors made all data underlying the findings in their manuscript fully available?

Reviewer #2: Yes

5. Is the manuscript presented in an intelligible fashion and written in standard English?

Reviewer #2: Yes

Reviewer #2: Many thanks to the authors for addressing my comments.

Only one final comment as I don't think the manuscript follows the journal guideline of 3500 words and maximum of 30 references. However, the editor will need to make the final decision regarding formatting.

**Do you want your identity to be public for this peer review?** For information about this choice, including consent withdrawal, please see our For information about this choice, including consent withdrawal, please see our Privacy Policy .

Reviewer #2: **Yes:** Ann-Beth MollerAnn-Beth Moller

---

## [Editor Report · Acceptance letter]

PONE-D-24-52959R3

PLOS One

Dear Dr. Habbous,

I'm pleased to inform you that your manuscript has been deemed suitable for publication in PLOS One. Congratulations! Your manuscript is now being handed over to our production team.

Kind regards,

on behalf of

Dr. Hale Teka

Academic Editor

PLOS One